# MPXGAT: An Attention based Deep Learning Model for Multiplex Graphs Embedding

## Abstract

Graph representation learning is a research area that has attracted a lot of attention in recent years. However, most of the existing studies focus on the embedding of single-layer graphs, which cannot describe systems where multiple types of relationships exist. Here we propose MPXGAT, an attention-based deep learning model for embedding multiplex graphs. Our methodology, which is based on GATs, embeds the nodes of a multiplex network by exploiting both their intra-layer and inter-layer connections, enabling link prediction tasks within and across different layers. A thorough experimental analysis on three benchmark datasets, shows that MPXGAT outperforms state-of-the-art competing algorithms.

## 1 Introduction and Related Works

In the last decades, graphs have proved to be a fundamental mathematical tool to model a variety of real-world complex systems. From transportation systems to power grids, from the network of our social relationships to that of neurons in our brain, complex networks are all around us. Due to such a ubiquity, network and graph theory has imposed itself in various research fields, from engineering to physics, from social science to biology (Newman, 2003; Boccaletti et al., 2006; Latora et al., 2017; Newman, 2018).

A topic that has recently received considerable interest in computer science is that of how efficiently represent large-scale graphs (Bacciu et al., 2020; Chen et al., 2020; Grohe, 2020). Particularly, graph embedding methods, which consist in projecting the elements of a graph, i.e., vertices, edges and motifs, to a low-dimensional vector space by preserving some of the graph properties, have shown to be very successful in graph representation (Khoshraftar & An, 2022). These embedding techniques are suitable for multiple applications, as they can be used in downstream learning tasks, including node classification (Rong et al., 2019), link prediction (Lü & Zhou, 2011), and community detection (Fortunato, 2010).

Graph embedding methods can be broadly categorized into traditional graph embedding and graph neural networks (GNNs) based graph embedding methods (Khoshraftar & An, 2022). The first group consists of algorithms that represent graphs relying on techniques such as random walks (Perozzi et al., 2014; Tang et al., 2015; Grover & Leskovec, 2016) and matrix factorization methods. These approaches, also known as shallow embedding methods, suffer from a series of drawbacks. Shallow approaches have several drawbacks that limit their efficiency and effectiveness. First, they do not share any parameters between the nodes in the encoder function, which maps each node to a vector representation. This makes them statistically and computationally inefficient. Second, they ignore the node attributes during the encoding process, which reduces the quality of the embeddings. Third, they are transductive, meaning that they can only generate embeddings for the nodes that were seen during the training phase (Hamilton et al., 2017b).

Embedding methods based on GNNs attempt to go beyond such limitations using more sophisticated encoders accounting for the graph structure and the node attributes. The key feature of a GNN is that the embedding of a node in the graph is obtained by *aggregating* the embeddings of the node's neighbors (Khoshraftar & An, 2022). In the last few years, a large variety of methods have been developed (Chen et al., 2020; Veličković et al., 2017; Liu et al., 2020; Gao & Ji, 2019), with the most notable examples including Graph Convolutional Networks (GCNs) (Kipf & Welling, 2016), Graph-SAGE (Hamilton et al., 2017a) and Graph Attention networks (GATs) (Veličković et al., 2017).

Despite their applicability, graphs present a limitation. Indeed, in their simplest form, graphs consist in a set of nodes and edges and they are able to capture only a single type of relationships among the system units (Latora et al., 2017). However, in many cases, the entities constituting a complex system may be connected through a variety of relationships. For instance, in social systems, same individuals can be connected through friendship, kinship or collaboration relationships and can communicate through different channels, including face-to-face interactions, phone calls and online social networks (OSNs) (Szell et al., 2010; Battiston et al., 2014). In biological systems, same proteins are characterized by genetic and physical interactions, as well as spatial co-localization (De Domenico et al., 2015b). To represent these and other systems, one can consider more complex mathematical structures (Boccaletti et al., 2014), such as multidimensional (Berlingerio et al., 2011) and multiplex graphs (Cozzo et al., 2018). The first consist in a set of vertices connected by edges with different labels, i.e., an edge-labelled multigraph, each representing a different type of interactions among the units of the system. The second, instead, is made up by a set of layers, each modeling a specific kind of relationships, that are interconnected with one another (De Domenico et al., 2013; Battiston et al., 2014). In multiplex graphs, two types of connections exist, namely the intra-layer links, connecting nodes on the same layer, and the inter-layer links, connecting instead nodes across different layers.

Multi-relational network embedding is a challenging problem garnering significant research interest in recent years. Various methods have been proposed to embed networks that have multiple types of nodes and links, such as multidimensional networks and multiplex networks. Some of these methods are based on shallow embedding approaches (Liu et al., 2017; Zhang et al., 2018; Shi et al., 2018; Gong et al., 2020), while others use graph convolutional networks (GCNs) (Yang et al., 2020; Ioannidis et al., 2020; Huang et al., 2020) or graph attention networks (GATs) (Cen et al., 2019). However, none of these methods can solve the problem of predicting links between different layers of a multilayer network. This problem is important when the network structure is incomplete or heterogeneous. For example, some methods assume that all nodes are present in every layer, which is not realistic in many cases. Other methods (such as (Behrouz & Seltzer, 2022); (Behrouz & Hashemi, 2022); (Qu et al., 2017a)) do not distinguish between intra-layer and inter-layer links, which ignores the diversity and complexity of multilayer networks. Therefore, there is a need for new methods that can address the inter-layer link prediction problem in a more general and flexible way. Very recently, MultiplexSAGE, a generalization of GraphSAGE aimed at embedding multiplex networks by relaxing these hypothesis, have been proposed (Gallo et al., 2023).

The inter-layer link prediction problem finds many crucial applications such as: i) in online social network (OSN) analysis, the linkage of user identities across different OSNs, an emerging task in social media that has attracted increasing attention in recent years (Shu et al., 2017). User identity linkage finds potential impact in different domains, from recommendation systems to cybersecurity (Tang et al., 2020); ii) identifying the same genes or proteins across different biological networks, such as gene expression, protein interaction, metabolic pathways, etc. This can help to discover the molecular mechanisms of diseases and to find potential drug targets (Jain et al., 2023); iii) matching the same entities across different knowledge graphs (Azmy et al., 2019). This can help to enrich the semantic information and to improve the query answering and reasoning capabilities.

The range of possible applications of inter-layer link prediction ultimately motivates the development of embedding algorithms for multiplex networks that are able to distinguish intra-layer and inter-layer links and to reconstruct both connectivity patterns.

We remark that predicting inter-layer connections in multiplex networks is related but not equivalent to either the graph matching problem or the graph alignment problem. Indeed, the goal of unlabeled graph matching is to identify an isomorphism between two graphs (Mathon, 1979). Therefore, the topology of two isomorphic graphs has to be exactly the same. Instead, the aim of inter-layer prediction is to find missing links between the same nodes in different layers of a multiplex network, that have in general different connectivity patterns. Concerning the global graph alignment (Ma & Liao, 2020), the goal is to find the best correspondences between two or more graphs by taking into account also the label of the nodes. The alignment task may lead to multiple solutions, including the one desired for the inter-layer link prediction, as various alignments can be equally good. Therefore, the inter-layer link prediction problem differs, as its objective is to find a given bijection between two graphs, i.e., the layers of the multiplex network. Moreover, graph alignment is a more general problem and can be applied to any graphs, while inter-layer prediction is a problem specific to multiplex networks.

In this paper, we introduce MPXGAT, an attention based deep learning model for multiplex graphs embedding. Our methodology, based on GATs, consists in embedding the nodes of a multiplex network by leveraging the information about their intra-layer and inter-layer connections, allowing for link prediction tasks both withing the same layer and cross different layers. We carry out a through experimental analysis on three benchmark datasets, showing that MPXGAT out-performs state-of-the-art competing algorithms. We conclude with an in-depth study of the model main features, proving how their use positively impacts the performance of the algorithm itself.

## 2 METHODS

In this section, we provide details of our proposed model, *MPXGAT*, which can embed multiplex networks, namely networks where multiple types of links exist. We first provide some preliminary notions, next we introduce the mathematical formulation of the model, and finally we describe the algorithmic implementation.

### 2.1 PRELIMINARY NOTIONS

We will start by defining the basic concepts of simple and multiplex graphs, then we will introduce the notation used in this work. A graph (network) is a mathematical structure that consists of a set of vertices (nodes) and a set of edges (links), where each edge represent a relation between a pair of vertices. However, simple graphs are not able to characterize systems where multiple types of interactions coexist, as they are able to capture only one kind of relationships. This need of a higher degree of expressiveness leads the definition multi-relational graphs, usually divided in two sub-groups, namely heterogeneous and multiplex graphs. In this work we focus on multiplex graphs, which can be defined as a graphs composed by different sub-networks which we refer here as horizontal and vertical. The horizontal network consists of a collection of simple graphs, called (horizontal) layers. In this setup, each node can be present in one or more layers, with each layer referring to a certain type of relationships We refer to the connections within a given layer as intra-layer links. The second sub-network, namely the vertical network, consists of a single-layer graph formed by the set of edges connecting nodes across different layers. We assume that a node $i$ on a layer $\alpha$ can be connected to at most one node $j$ on another layer $\beta$, i.e., the two nodes represent the same unit of the system. Also, we assume and that if a node $i$ on layer $\alpha$ is connected to a node $j$ on layer $\beta$, and if $j$ is connected to a node $k$ on layer $\gamma$, then nodes $i$ and $k$ are also connected. Under these hypothesis, the vertical network consists in a collection of different connected components, which can be either cliques or isolated nodes. The edges in the vertical network will be referred to as inter-layer links.

Mathematically, a multiplex graph is a set $\mathbb{V}$ of $N$ nodes that are connected through $L$ different layers. We assume that $N_\alpha$ nodes are present in each layer $\alpha \in \{0, 1, \ldots, L\}$, such that $N_1 + \cdots + N_L = N$. Each layer $\alpha$ is a graph $\mathcal{G}_\alpha = (\mathbb{V}_\alpha, \mathcal{E}_\alpha)$ where $\mathbb{V}_\alpha \in \mathbb{V}$ is the set of the $N_\alpha$ nodes and $\mathcal{E}_\alpha$ is the set of edges among them. The node sets for each layer are disjoint, i.e., $\mathbb{V}_\alpha \cap \mathbb{V}_\beta = \emptyset$ for $\alpha \neq \beta$. The set of intra-layer links for the horizontal network can be defined as $\mathcal{E}_{intra} = \bigcup_{\alpha=1}^{L} \mathcal{E}_\alpha$. The vertical network can be instead defined as $\mathcal{G}_V = (\mathbb{V}, \mathcal{E}_{inter})$, where $\mathcal{E}_{inter} = \{(i, j) \in \mathbb{V} \times \mathbb{V} \mid \exists \alpha, \beta \in \{0, 1, \ldots, L\} s.t. (i, j) \in \mathcal{E}_\alpha \times \mathcal{E}_\beta\}$ is the set of inter-layer edges. An example of a multiplex network is reported in Figure 1.

From now on, we will use the superscript $\cdot^H$ when we refer to the horizontal network, while the superscript $\cdot^V$ will refer to the vertical one.

In Table 1 we report the notation, together with the most relevant variables used in our model. Note that, in some definitions, superscripts $\cdot^H$ and $\cdot^V$ are omitted for the sake of readability and to generalize those concepts to both horizontal and vertical networks.

### 2.2 THE MPXGAT GENERAL FRAMEWORK

The main idea of MPXGAT is to generate two separate embeddings for each node in two different phases. In the first phase, each node is embedded according to the horizontal layers where it has multiple types of relations with other nodes. In the second phase, nodes are embedded according to

Table 1: Description of the variables involved in this work.

| VARIABLE | DESCRIPTION |
| --- | --- |
| $N \in \mathbb{N}$ | Number of nodes in the multiplex graph |
| $\mathbb{V}$ | Set of nodes in the multiplex graph |
| $L \in \mathbb{N}$ | Total number of horizontal layers of the multiplex graph |
| $\alpha \in \mathbb{N}$ | Index of the horizontal layer in the multiplex graph |
| $N_\alpha$ | Number of nodes within the horizontal layer of index $\alpha$ |
| $\mathbb{V}_\alpha$ | Set of nodes within the horizontal layer of index $\alpha$ |
| $\mathcal{E}_\alpha$ | Set of edges within the horizontal layer $\alpha$ |
| $\mathcal{G}_\alpha$ | Layer of index $\alpha$ in the horizontal network |
| $\mathcal{E}_{intra}$ | Set of intra-layer edges for the horizontal network |
| $\mathcal{E}_{inter}$ | Set of inter-layer edges for the horizontal network |
| $\mathcal{G}_V$ | Vertical network of the multiplex graph |
| $F \in \mathbb{N}$ | Initial node feature dimensionality |
| $F' \in \mathbb{N}$ | Final node feature dimensionality |
| $i \in \mathbb{N}$ | Source node |
| $j \in \mathbb{N}$ | Destination node |
| $k \in \mathbb{N}$ | Source node horizontal layer's index |
| $q \in \mathbb{N}$ | Destination node horizontal layer's index |
| $\mathcal{N}_i$ | Set of nodes connected to node $i$ |
| $\boldsymbol{h}_i \in \mathbb{R}^F$ | Embedding of node $i$ |
| $\boldsymbol{v} \in \mathbb{R}^{F'}$ | Attention weight vector |
| $\boldsymbol{W} \in \mathbb{R}^{F \times F'}$ | Weight matrix, used to linearly transform the node embeddings |
| $e_{i,j} \in \mathbb{R}$ | Attention coefficient between node $i$ and $j$ |
| $\alpha_{i,j} \in \mathbb{R}$ | Normalized attention coefficient between nodes $i$ and $j$ |
| $\boldsymbol{h}_i \in \mathbb{R}^{F'}$ | Updated embedding of node $i$ after a forward pass |
| $\sigma$ | Activation function (in our case LeakyReLU) |
| $f$ | Function used to transform the horizontal embeddings to the same dimensional space of the vertical embedding |
| $g$ | Function used to combine the data obtained from the vertical network with the result of the application of the function $f$ |

the vertical network where they are linked to their counterparts on different layer. The structure of the model is reported in Figure 2.

The embedding algorithm uses two sub-models: MPXGAT-H and MPXGAT-V. MPXGAT-H applies a series of GAT convolutional layers independently to each layer of the horizontal network, while MPXGAT-V implements a series of modified GAT convolutional layers, called GAT-V, to the vertical one. Concerning each layer of the horizontal network, the embedding is described by the

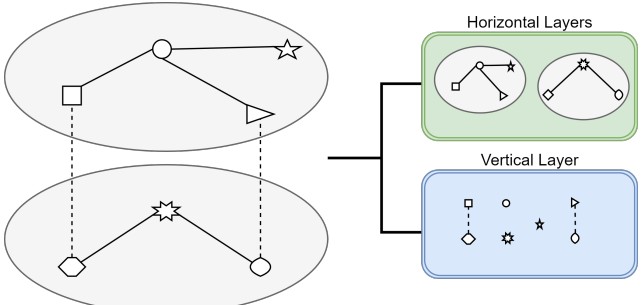

Figure 1: A toy example of a multiplex network with 2 horizontal layers. The solid edges represent the intra-layer connections while the dashed edges are the inter-layer edges.

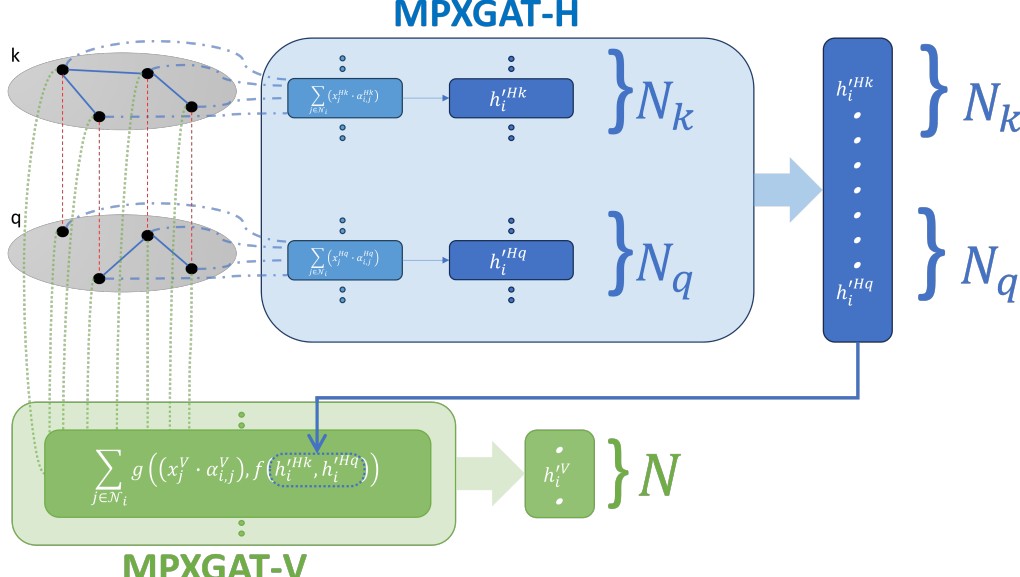

Figure 2: The structure of the MPXGAT model. In this toy example, it is applied on a multiplex network with 2 horizontal layers where the solid blue edges represent the intra-layer connections while the dashed red edges are the inter-layer edges. The data is provided to the *MPXGAT-H* throughout the dot-and-dash blue lines. Once processed these are used to feed the *MPXGAT-V* together with the inter-layer links (the dotted green lines). The output of the model consists of both horizontal and vertical nodes embedding.

following equations:

$$e_{i,j}^{H_k} = \text{LeakyReLU}\left(\left[\boldsymbol{W}^{H_k} \cdot \boldsymbol{h}_i^{H_k} \cdot \boldsymbol{v}^{H_k T} || \boldsymbol{W}^{H_k} \cdot \boldsymbol{h}_j^{H_k} \cdot \boldsymbol{v}^{H_k T}\right]\right) \tag{1a}$$

$$\alpha_{i,j}^{H_k} = \text{softmax}(e_{i,j}^{H_k}) = \frac{\exp(e_{i,j}^{H_k})}{\sum_{z \in \mathcal{N}_i^{H_k}} \exp(e_{i,z}^{H_k})} \tag{1b}$$

$$\boldsymbol{h}_i^{'H_k} = \sigma\left(\sum_{j \in \mathcal{N}_i^{H_k}} \alpha_{i,j}^{H_k} \cdot \boldsymbol{W}^{H_k} \cdot \boldsymbol{h}_j^{H_k}\right) \tag{1c}$$

Equations 1 use the convolutional layer as described in the GAT model (Veličković et al., 2017). In Equation 1c, $\boldsymbol{h}_i^{'H_k}$ represents the horizontal embedding of the node $i$ that belongs to the layer $k$, while in Equation 1b $\alpha$ represents the self attention mechanism.

The horizontal embeddings are then employed in the generation of the vertical embeddings by making use of a custom mechanism inspired by the one applied in the GAT model. The equations to compute the vertical embedding of node $i$ are the following:

$$e_{i,j}^{V^{k,q}} = \text{LeakyReLU}\left(\left[\boldsymbol{W}^V \cdot \boldsymbol{h}_i^V \cdot \boldsymbol{v}^{VT} || \boldsymbol{W}^V \cdot \boldsymbol{h}_j^V \cdot \boldsymbol{v}^{VT}\right]\right) \tag{2a}$$

$$\alpha_{i,j}^{V^{k,q}} = \text{softmax}(e_{i,j}^{V^{k,q}}) = \frac{\exp(e_{i,j}^{V^{k,q}})}{\sum_{z \in \mathcal{N}_i^V} \exp(e_{i,z}^V)} \tag{2b}$$

$$\boldsymbol{h}_i^{'V} = \sigma\left(\sum_{j \in \mathcal{N}_i^V} g\left(\left[\alpha_{i,j}^V \cdot \boldsymbol{W}^V \cdot \boldsymbol{h}_j^V\right], f(\boldsymbol{h}_i^{H_k}, \boldsymbol{h}_j^{H_q})\right)\right) \tag{2c}$$

Equations 2 introduce two new elements. The first is the function $f$, which is applied to the final horizontal embeddings and transforms them to the same dimensional space of the vertical embeddings. This is guided by the linear transformation given by the matrix $\boldsymbol{W}^V$. Such a function can be a simple linear transformation of the node horizontal embeddings, or it can be a more complex function adapted to the specific task considered. The second generalization introduced is the usage of a function $g$, which combines the data obtained from the current vertical embedding with the results of the the function $f$.

## 2.3 MPXGAT IMPLEMENTATION FOR LINK PREDICTION

Here, we present an implementation of *MPXGAT* that is suitable for link prediction.

A set of customization are applied to both *MPXGAT-H* and *MPXGAT-V*'s attention mechanisms, described in Equations 1a, 1b and 2a, 2b, respectively. In particular we have:

- in Equations 1a and 2a, in place of a single weight matrix for both $i$ and $j$ nodes, two different weight matrices, namely $(\boldsymbol{W}_i, \boldsymbol{W}_j)$, are used. This improves the representation capabilities of the model, at the expense of a greater number of parameters. The same choice is done for the weight vector by introducing $(\boldsymbol{v}_i, \boldsymbol{v}_j)$ vectors. To improve the generalization capabilities of the model, the concatenation $||$ operator is implemented as a sum $+$ and both parts of this operation are augmented with the introduction of a bias vector;

- in Equations 1b and 2b to grant to the model a better generalization capability, an additional dropout mechanism is applied to the attention coefficients, so to sub sample the paths present in the graph;

Another addition is the usage of *multiple attention heads*, which consist of calculating multiple embeddings of the same nodes by using different applications of the convolutional layer. This creates multiple representations that highlight different kinds of patterns about the nodes, since every attention head can be seen as an information channel representing a certain aspect of nodes. The implemented sub-models concatenates the embeddings for each attention head in the hidden convolutional layers of the model and uses a mean operation for the final one. Finally, for both sub-models the activation function $\sigma$ in Equations 1c and 2c is the *LeakyReLU*.

The general *MPXGAT* framework uses generic $f$ and $g$ functions for the embedding, which can be specified according to the particular task we want to solve. Here, the two functions $f$ and $g$ introduced in Equation 2c have the following characteristics:

- $f$ use a linear transformation of the horizontal embedding of the source node, $\boldsymbol{h}_i^H$, to the same dimensional space of its vertical embedding, $\boldsymbol{h}_i^V$, here described with $\boldsymbol{x}_i^H$. This is done by using an additional weight matrix $\boldsymbol{Z}^H$. We then multiply the result by a vector of parameters $\boldsymbol{v}_i^H$, later applying the LeakyReLU function. Here, $\boldsymbol{v}_i^H$ has the role of enhancing the useful patterns from the horizontal embeddings.

- The function $g$ performs a weighted sum of the output of the function $f$ (see Equation 3c) and the data computed with the convolutional layer for the vertical network (see Equation 3d). The scalar parameter $\beta$, automatically inferreed during the learning process, is introduced to balance the information coming from the horizontal and vertical embeddings, respectively. In particular when $\beta = 0$ the horizontal embedding is not considered, while for $\beta = 0.5$ both the components are considered with the same weight.

  Finally, the equations describing the implementation of *MPXGAT* for link predictions are:

$$x_i^H = h_i^H \cdot Z^H + b_{h_i} \tag{3a}$$

$$\alpha_i^H = \text{LeakyReLU}\left( x_i^H \cdot v_i^H \right) \tag{3b}$$

$$m_i^H = f(x_i^H) = \alpha_i^H * x_i^H \tag{3c}$$

$$m_{i,j}^V = \alpha_{i,j}^V \cdot x_j^V \tag{3d}$$

$$g\left(m^V, m^H\right) = \left(m_{i,j}^V * (1 - \text{ReLU}\,(\beta))\right) + \left(\left(m_i^H\right) * \text{ReLU}\,(\beta)\right) \tag{3e}$$

From our model we can notice that the constraint of having the same number of nodes in each layer does not apply in our case, as each horizontal layer can have a different number of nodes. A second aspect that highlights the flexibility of the model is the possibility to support node features and edge weights. Indeed different features can be used by the two sub-models, since the horizontal and the vertical networks are defined and processed as separated entities. As we will report in the experimental section, this characteristic guarantee good performances of the *MPXGAT* model even in datasets where the edge density in the horizontal or vertical networks is high, which showed to be a limit of algorithms based on GraphSAGE (Gallo et al., 2023).

## 3  RESULTS

In this section, we present the results obtained by *MPXGAT* regarding the prediction of both intra-layer and inter-layer links. We first present the datasets studied, as well as the experimental setup used for the analysis and the competing methods considered as benchmarks for our algorithm. Then, we test the performances of *MPXGAT* in predicting the links of different multiplex networks (the source code of the model is provided as supplementary materials). We round up the analysis by assessing the impact of the horizontal and vertical submodels on the embedding procedure.

### 3.1  DATASET

To assess the performance of *MPXGAT* in predicting both intra-layer and inter-layer connections, we employed the same datasets that have been originally used to test *MultiplexSAGE* (Gallo et al., 2023). These include data relative to three types of real-world multiplex networks, namely a collaboration, a biological, and an online social networks. Specifically, we considered the following datasets.

**arXiv** (De Domenico et al., 2015a). The arXiv multiplex network represents collaborations in various research topics published on the pre-print archive. Each layer of the network corresponds to a different research category or theme. The network was obtained selecting papers published before May 2014 that contain the keyword *networks* in their titles or abstracts.

**Drosophila** (Stark et al., 2006). This multiplex network represents the interactions between proteins and genes in the common fruit fly, i.e., Drosophila melanogaster, with each layer representing interactions of various types. The dataset was collected from the Biological General Repository for Interaction Datasets (BioGRID), with data updated until January 2014.

**ff-tt-yt** (Dickison et al., 2016). This multiplex network is derived from Friendfeed (ff), a social media aggregation platform where users can link their accounts from various online social networks (OSNs). The network comprises users who have registered a single Twitter (tw) account and a sole YouTube (yt) account on Friendfeed. Additionally, the Twitter and YouTube accounts are linked to a single Friendfeed account.

For each empirical dataset, we consider exclusively the largest connected component of the multiplex networks and we treated all networks as undirected and unweighted. Details about the largest connected component for each of these datasets are provided in Table 2.

We remark that, for all datasets considered in our analysis, nodes are not provided with any external features. Hence, we associate to each node, $n$, a one-hot encoding vector defined by the Kronecker function, $\delta_{i,n}$.

Table 2: Dataset Information including the number of nodes, edges, and average degree for the largest connected component of each dataset (arXiv, Drosophila, and ff-tw-yt)

| Dataset | Nodes | Edges | Avg. Degree |
|---------|-------|-------|-------------|
| arXiv | 19,310 | 20,738 | 1.07 |
| Drosophila | 11,867 | 5,171 | 0.44 |
| ff-tw-yt | 11,827 | 6,028 | 0.51 |

## 3.2 EXPERIMENTAL SETUP

To assess the performance of *MPXGAT*, we consider the experimental setup delineated in Gallo et al. (2023). We partition the data following a multiple step procedure. First, we randomly select 20% of the network nodes, labeling them as *marked nodes*. We then define test and training sets. Both sets encompass positive and negative examples, the former representing actual links within the network, while the latter consisting in pairs of unconnected nodes. In the test set, positive examples comprise a subset of 20% of the intra-layer links and all inter-layer links among the marked nodes. Conversely, positive examples in the training set include all remaining intra-layer and inter-layer links in the multiplex network. As negative examples within the test set, we include 20% of all possible negative intra-layer links among the marked nodes, as well as all possible inter-layer links between them. The negative examples in the training set are constituted by the remaining pairs of unconnected nodes.

To test the performance of our algorithm, instead of conducting a single experiment for each dataset, we repeat the embedding procedure multiple times. For each repetition, we randomly select a subset of marked nodes, defining the training and test sets accordingly. To obtain the best parameter settings, we applied the grid search method.

## 3.3 COMPETING METHODS

We conduct a comparative analysis of *MPXGAT* against three competing methodologies, specifically *GraphSAGE*, *GATNE* and *MultiplexSAGE*.

***GraphSAGE*** (Hamilton et al., 2017a). *GraphSAGE* is an inductive node embedding algorithm that leverages node features to learn an embedding function capable of generalizing to unseen nodes. It was originally designed for single-layer network embeddings, so in our experiments, we apply it without distinguishing between intra-layer and inter-layer links.

***GATNE*** (Cen et al., 2019). *GATNE* is an embedding algorithm for attributed multiplex heterogeneous networks, encompassing multigraphs with diverse node and edge types. To adapt *GATNE* to our specific task, we introduced two categories of edges, denoted as intra-layer and inter-layer links. This adjustment allows us to create a multigraph that the algorithm can learn to embed effectively.

***MultiplexSAGE*** (Gallo et al., 2023). *MultiplexSAGE* represents an extension of the GraphSAGE algorithm, specifically tailored for embedding multiplex networks. Its key features is the distinction made between inter-layer and intra-layer links that allows for the prediction of both intra-layer and inter-layer connectivity patterns.

Other relevant methods such as (Gong et al., 2020), (Ioannidis et al., 2020), (Liu et al., 2017), (Qu et al., 2017b), (Zhang et al., 2018), (Shi et al., 2018) and (Huang et al., 2020) can not be used as competing methods, as they assume that each layer has the same number of nodes and that all inter-layer connections are known. Therefore, as we are interested in predicting both intra-layer and inter-layer links, these methods are not suitable benchmarks for evaluating the performance of *MPXGAT*.

## 3.4 EMBEDDING MULTIPLEX NETWORKS

Our first analysis concerns the prediction of both intra-layer and inter-layer links when embedding a multiplex network. For each embedding, we evaluate the Area Under the Receiveing Operating Characteristic (ROC) Curve (AUC), and we take an average over the different repetitions as a performance metric. We consider the standard deviation as an indicator of statistical error. Table 3 provides an overview of the performances obtained with *MPXGAT*, *GraphSAGE*, *GATNE*, and *MultiplexSAGE*. We note that *GATNE* outperforms competitors in intra-layer link prediction, having a higher average AUC for all three datasets. However, *MPXGAT* has comparable performances with *GATNE* on the ff-ww-tt and the Drosophila dataset, outperforming both *GraphSAGE* and *MultiplexSAGE* on the task. Concerning the inter-layer link prediction, *MPXGAT* clearly performs better than all other algorithms, including *MultiplexSAGE*, which is explicitly designed for that task. If we consider instead the general behavior of the methods without distinguishing intra- and inter-layer connections, our algorithm is clearly the best solution, as reported in Table 4.

Table 3: Performance comparison on intra-layer and inter-layer link prediction across the three distinct datasets: ff-tw-yt, Drosophila, and arXiv. The assessment is based on the AUC metric, using the standard deviation as error metric. The best performing tool is highlighted in boldface

| Algorithm | ff-tw-yt | | Drosophila | | arXiv | |
|---|---|---|---|---|---|---|
| | *intra* | *inter* | *intra* | *inter* | *intra* | *inter* |
| *GraphSAGE* | 0.47 (± 0.02) | 0.56 (± 0.02) | 0.54 (± 0.02) | 0.63 (± 0.02) | 0.72 (± 0.02) | 0.70 (± 0.01) |
| *GATNE* | **0.83 (± 0.01)** | 0.47 (± 0.01) | **0.78 (± 0.01)** | 0.55 (± 0.01) | **0.91 (± 0.01)** | 0.63 (± 0.01) |
| *MultiplexSAGE* | 0.48 (± 0.02) | 0.62 (± 0.02) | 0.51 (± 0.01) | 0.77 (± 0.02) | 0.71 (± 0.02) | 0.83 (± 0.01) |
| *MPXGAT* | 0.76 (± 0.06) | **0.83 (± 0.01)** | 0.76 (± 0.05) | **0.86 (± 0.02)** | 0.80 (± 0.02) | **0.84 (± 0.01)** |

Table 4: Overall AUC performaces calculated as a weighted sums of the results shown in Table 3, based on the number of edges used to evaluate the models.

| Algorithm | ff-tw-yt | Drosophila | arXiv |
|---|---|---|---|
| *GraphSAGE* | 0.49 (± 0.02) | 0.57 (± 0.01) | 0.70 (± 0.01) |
| *GATNE* | 0.72 (± 0.01) | 0.69 (± 0.02) | 0.72 (± 0.01) |
| *MultiplexSAGE* | 0.52 (± 0.02) | 0.61 (± 0.01) | 0.79 (± 0.01) |
| *MPXGAT* | **0.78 (± 0.03)** | **0.80 (± 0.03)** | **0.82 (± 0.04)** |

As described in Section 2, the better performance of *MPXGAT* in predicting inter-layer links stems from the use of two distinct embeddings, dealing with the horizontal, i.e., intra-layer, and vertical, i.e., inter-layer, embeddings, respectively. In contrast, the other models rely on a single embedding that serves both tasks. As we will further investigate in the next section, this architectural difference allows *MPXGAT* to predict with a certain reliability both intra-layer and inter-layer links.

## 3.5 MEASURE THE IMPACT OF HORIZONTAL EMBEDDINGS

We now establish the impact of the horizontal embeddings on the performance of *MPXGAT*. To do so, we conduct two experiments. In the first, we perform the embedding with *MPXGAT*, but instead of using MPXGAT-V, which relies on the horizontal embeddings generated by MPXGAT-H, for embedding the vertical network, we consider a standard GAT (Veličković et al., 2017), thus ignoring the contribution of MPXGAT-H. Table 5 reports the average AUC for the inter-layer link prediction in both configurations. We observe that ignoring the horizontal embedding leads to worse performance across all datasets, suggesting that including the MPXGAT-V submodel increases the algorithm adaptability and predictive power.

To validate the significance of these findings we performed a Welch's T-test. The p-values are $6.00 \times 10^{-7}$, $9.10 \times 10^{-10}$, and $1.10 \times 10^{-7}$ for the Drosophila, arXiv, and ff-tw-yt datasets, respectively, confirming the statistical significance of our finding.

In the second experiment, we keep the original model architecture but instead of providing MPXGAT-V with the horizontal embeddings generated by MPXGAT-H, we replace them with random embeddings, i.e., vectors whose components are random values.

Table 5: Results of the ongoing experiment aimed at assessing the impact of excluding horizontal embeddings on inter-layer link prediction. The performance is evaluated in terms of AUC across the three datasets. The second model, which omits horizontal embeddings, exhibits lower performance in inter-layer link prediction across all datasets.

| Algorithm | ff-tw-yt | Drosophila | arXiv |
|---|---|---|---|
| *MPXGAT-V(layer* GAT-V*)* | **0.83 ± (0.01)** | **0.86 ± (0.01)** | **0.84 ± (0.01)** |
| *GAT(layer* GAT*)* | 0.72 ± (0.02) | 0.78 ± (0.02) | 0.78 ± (0.01) |

Table 6 shows the results of the comparison with regards to the inter-layer link prediction. We note that for both the ff-tw-yt and the arXiv dataset, the usage of the horizontal embedding increases the performance of the prediction task, while we observe similar values of the average AUC for the Drosophila dataset. The statistical significance of this result is confirmed by the Welch's T-test, for which we obtain p-values equal to $6.10 \cdot 10^{-6}$, 0.75, $3.60 \cdot 10^{-4}$ for the arXiv, Drosophila, and ff-ww-tt datasets, respectively.

The Welch's test confirms that, for the Drosophila dataset, there is no significant difference in the performance between using the horizontal embeddings generated by MPXGAT-H and using random embeddings instead.

We conjecture that this outcome is due to the structure of the multiplex network, both in terms of the size of the different layers, i.e., the number of nodes laying on them, and of the connectivity patterns, both within the same layer (e.g., randomness, clustering and community organization, and across different ones, i.e., overlapping).

Table 6: Results of the experiment investigating the impact of replacing meaningful horizontal embeddings with random embeddings. The performance is evaluated in terms of AUC. The model with random embeddings performs worse in two out of the three datasets compared to the model with actual embeddings, indicating a significant decrease in predictive accuracy.

| Algorithm | ff-tw-yt | Drosophila | arXiv |
|---|---|---|---|
| *MPXGAT-V(actual embedding)* | **0.83 ± (0.01)** | 0.86 ± (0.01) | **0.84 ± (0.01)** |
| *MPXGAT-V(random embedding)* | 0.80 ± (0.02) | 0.86 ± (0.01) | 0.81 ± (0.01) |

## 4 CONCLUSIONS

In this paper we have introduced *MPXGAT*, an attention based deep learning model for the embedding of multiplex graphs. Through a comprehensive experimental analysis we showed that MPXGAT out-performs state-of-the-art competing algorithms. Future work will be aimed at better understanding how the community structure withing each layer influences the performances of the algorithm in terms of the reliability of predicted inter-layer relations.

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
