# OpenReview forum: "MPXGAT: An Attention based Deep Learning Model for Multiplex Graphs Embedding"
_ICLR.cc/2024/Conference — Submitted to ICLR 2024_

### Official Review · Reviewer_x5r2 · 2023-10-31

**Soundness:** 2 fair
**Presentation:** 2 fair
**Contribution:** 2 fair
**Rating:** 3
**Confidence:** 5

**Summary:**

The author propose a node embedding method based on single-layer GAT on multiplex graphs. The research problem is driven from real-world application. The proposed approach consists of two set of node information aggregation. One is from intra-layer node neighbor and the other is to fuse cross-graph-layer connections. Some existing datasets and multiplex graph-embedding baselines are involved in experiments.

**Strengths:**

S1. The setup of multiplex graph are from real-would applications

S2. The method is easy to understand in general

**Weaknesses:**

W1. Methodology is lack of clear description

W2. Unclear representation, figure with toy examples could help

W3. Experimental results are not convincing, not enough baseline, no clear analyses for the results

**Questions:**

D1 Unclear description

Definition of vertical network (Page-3), "consists of a single-layer graph formed by the set of edges connecting nodes across different layers". Is it for a single layer or multiple layers?

"We assume that a node i on a layer /alpha can be connected to at most one node j on another layer /beta, i.e., the two nodes represent the same unit of the system" do the authors mean node i and node j is the same node?

"N1 +···+ NL = N" N_i and N_j may have overlapping nodes, right?

Does horizontal network for multiple edge types for a set to nodes and vertical network is a set of cross edges among different layers. I have research experience on the graph construction but I am almost lost by the description. It's recommended to illustrate the abstract definition with a toy graph or from a real-would example. For example two people can build relationships via multiple social media, e.g. Facebook, Twitter, YouTube, TikTok... Then explain where are the horizontal and vertical subnetworks in the example.


D2. "Equations 1 use the convolutional layer as described in the GAT model" (Page 3)? Does the convolutional layer refer to the unit block in GCN? One of the main difference between GCN and GAT is the aggregation mechanism (auto-weighted vs unweighed). Can GAT layer be considered as a convolutional layer? Any reference?


D3. An interesting question is that how many MPXGAT-H layers needed before forwarding node embedding to MPXGAT-V, and vice versa. The authors seems not mention the detail. Is the default as 1 MPXGAT-H -> 1 layer MPXGAT-V -> 1 layer MPXGAT-H...? Then how many rounds of H-V block needed?

D4. A simple baseline is applying a single-layer graph embedding method (e.g., DeepWalk, Node2vec, LINE, GCN, GAT, and etc) and concate them via multiple graph layers to do link prediction

D5. It's easy to observe that GATNE performs clearly better for intra-layer task than others (including the proposed one). But why? The authors doesn't analyze the reason

========================

I acknowledge that I have read the authors response. I appreciate the authors effort. But it didn't address my concerns. I would keep my original rating.

---

> ### Author Response · Authors · 2023-11-22
> **Reviewer 3 comments' answers**
>
> **Question 1**
>
> ***Author answer:***
>
> We thank the Referee for the valuable comments. We tried to improve the readability of our manuscript by clarifying the definitions in accordance with the suggestions provided. In particular, we have now added a graphical illustration of how our algorithm would represent a toy example of a multiplex network in terms of horizontal and vertical layers.
>
> ---
>
> **Question 2**
>
> ***Author answer:***
>
> We agree with the reviewer that a GAT layer can be considered convolutional since it also uses a message-passing mechanism between the nodes of the graph. The main difference between a standard GCN and a GAT layer is the aggregation method used, which in GAT is based on attention mechanisms, while this is not the case for GCN, as GCN uses a simple weighted average of linear applications of the messages.
>
> ---
>
> **Question 3**
>
> ***Author answer:***
>
> The MPXGAT model consists of two submodels: MPXGAT-H and MPXGAT-V. MPXGAT-H generates K embeddings for each node, one for each of the K horizontal networks (the layers of the multiplex network). While, MPXGAT-V uses these embeddings to compute another set of embeddings for each node, considering the vertical network. The final output consists of K+1 embeddings per node: the first K is used for intra-layer link prediction, and the last one is used for inter-layer link prediction. The MPXGAT model applies MPXGAT-H and MPXGAT-V only once, without any loops or iterations.
>
> ---
>
> **Question 4**
>
> ***Author answer:***
>
> The Referee is correct in stating that this can be another interesting baseline to consider. In our case, we have considered three baselines: one is a single-layer graph embedding method, i.e., GraphSAGE, as suggested by the Referee. The other two, i.e., GATNE and MultiplexSAGE, are instead algorithms developed to embed heterogeneous graphs and multiplex networks, respectively.
>
> ---
>
> **Question 5**
>
> ***Author answer:***
>
> GATNE performs the embedding on the whole graph joining the knowledge coming from the intra-layer connections and those from the inter-layers. Therefore by considering the global performance, there is a general gain against using a single GAT model for each layer as done in the MPXGAT-H submodel. We speculate that by applying GATNE on each layer and leveraging the obtained results, the outgoing results should be comparable to the MPXGAT-H model.
> We also remark that if we consider the general behavior of the methods without distinguishing intra- and inter-layer connections, one can see that our algorithm is clearly the best solution. Such an analysis is reported in Table 4 of our revised paper.

---

### Official Review · Reviewer_NRz2 · 2023-11-01

**Soundness:** 2 fair
**Presentation:** 3 good
**Contribution:** 2 fair
**Rating:** 3
**Confidence:** 4

**Summary:**

This paper introduces an attention-based graph encoding method based on GATs for encoding multiplex graphs. This approach allows considering  both their intra-layer and inter-layer connections in the node encoding, enabling the prediction of connections between different layers.

**Strengths:**

The paper is well written and the experiments show improvements in predicting inter-layer connections.

**Weaknesses:**

1. The motivation of the paper is unclear in the current version. In what applications do we need to predict inter-layer connections? Does predicting inter-layer connections in multiplex networks mean alignment of entities and so equivalent to graph alignment problem? Why link prediction in heterogeneous graphs cannot be used for predicting the inter-layer connections? That is, one can consider a multiplex network as a special case of heterogeneous graphs and then apply link prediction methods in heterogeneous graphs, which can both predict inter-layer and intra-layer links.

2. There is a lack of discussion with many important and relevant studies. For example [1, 2, 3] also use attention modules to learn the node encodings in multiplex graphs. How this GAT-based approach is different from these attention modules and how it can be compared with them? It would be better if the authors could provide a detailed discussion on this. In addition to these methods, as mentioned above, it would be better if the authors could discuss heterogeneous graph-learning methods [4, 5].

3.  The model design is a simple application of GAT on multiplex networks, and I believe the contribution to the model design is not novel and somehow incremental.

4.  The experiments show that GATNE, which is not state-of-the-art for link prediction in heterogeneous graphs, can outperform the proposed method in intra-layer link prediction. Based on this result, it seems that the only application of the proposed method is inter-layer link prediction, which has unclear motivation.






$ $
$ $

1. Anomaly Detection in Multiplex Dynamic Networks: from Blockchain Security to Brain Disease Prediction. NeurIPS TGL workshop 2022.
2. An attention-based collaboration framework for multi-view network representation learning. CIKM 2017.
3. CS-MLGCN: Multiplex graph convolutional networks for community search in multiplex networks. CIKM 2022.
4. A multi-view contrastive learning for heterogeneous network embedding.  Scientific Reports 2023.
5. Fast attributed multiplex heterogeneous network embedding. CIKM 2020.

**Questions:**

Please see the questions in the Weaknesses..

---

> ### Author Response · Authors · 2023-11-22
> **Reviewer 2 comments' (1,2,3) answers**
>
> **Question 1.**
>
> ***Author answer:***
>
> We thank the reviewer for raising these relevant aspects. We tried to clarify them through the text.  The inter-layer link prediction problem finds many crucial applications such as: i) in online social network (OSN) analysis, the linkage of user identities across different OSNs, an emerging task in social media that has attracted increasing attention in recent years (Shu et al., 2017). User identity linkage finds potential impact in different domains, from recommendation systems to cybersecurity (Tang et al., 2020); ii) identifying the same genes or proteins across different biological networks, such as gene expression, protein interaction, metabolic pathways, etc. This can help to discover the molecular mechanisms of diseases and to find potential drug targets (Jain et al., 2023); iii) matching the same entities across different knowledge graphs (Azmy et al., 2019). This can help to enrich the semantic information and to improve the query answering and reasoning capabilities. The range of possible applications of inter-layer link prediction ultimately motivates the development of embedding algorithms for multiplex networks that are able to distinguish intra-layer and inter-layer links and to reconstruct both connectivity patterns.
> We remark that predicting inter-layer connections in multiplex networks is related but not equivalent to either the graph matching problem or the graph alignment problem. Indeed, the goal of unlabeled graph matching is to identify an isomorphism between two graphs (Mathon, 1979). Therefore, the topology of two isomorphic graphs has to be exactly the same. Instead, the aim of inter-layer prediction is to find missing links between the same nodes in different layers of a multiplex network, that have in general different connectivity patterns. Concerning the global graph alignment (Ma & Liao, 2020), the goal is to find the best correspondences between two or more graphs by taking into account also the label of the nodes. The alignment task may lead to multiple solutions, including the one desired for the inter-layer link prediction, as various alignments can be equally good. Therefore, the inter-layer link prediction problem differs, as its objective is to find a given bijection between two graphs, i.e., the layers of the multiplex network. Moreover, graph alignment is a more general problem and can be applied to any graphs, while inter-layer prediction is a problem specific to multiplex networks.
> The Referee is also right in pointing out that one can use heterogeneous graph embedding algorithms to perform inter-layer link predictions by distinguishing two types of edges, i.e., intra-layer and inter-layer. This is actually how we run GATNE in our experiments. However, one can clearly see that MPXGAT outperforms such a method. However, we want to clarify that link prediction in heterogeneous graphs should not be directly used for predicting the inter-layer connections, because heterogeneous graphs have different types of nodes and edges, while multiplex networks have the same type of nodes but different types of edges. Link prediction in heterogeneous graphs requires to consider the node and edge attributes and semantics, while inter-layer prediction requires to consider the layer and network properties.
>
> ---
>
> **Question 2.**
>
> ***Author answer:***
>
> We thank the reviewer for pointing us to these relevant references which have been included in our paper.  However, we must point out that all these methods are not directly comparable with the MPXGAT model since the attention mechanism is being used in order to perform a weighted addition of the embeddings generated using their relative layer scope.
> Those models generate a single final embedding per node, in contrast to the K + 1 embeddings (where K is the number of layers of the multiplex network and the additional one is calculated for inter-layer link prediction).
> Also, we point out that these methods do not consider the inter-layer links since they are based on the assumption that every node is present in every layer and that the linkage among them is known. Yet, the revised version of the manuscript now includes these relevant references.
>
> ---
>
> **Question 3.**
>
> ***Author answer:***
>
> The Referee is right in pointing out that our contribution is based on the GAT model proposed by Veličković et al. to embed single layers. However,  MPXGAT represents a generalization of such a model to deal with the vertical layers of a multiplex network. Indeed, the vertical embedding submodel, namely MPXGAT-V uses a modified version of GAT. Therefore, we believe that our method can not be considered a simple application of GAT, in contrast with other very relevant contributions such as GATNE, which are more clearly based on GAT.

---

> > ### Author Response · Authors · 2023-11-22
> > **Reviewer 2 comment 4 answer**
> >
> > **Question 4**
> >
> > ***Author answer:***
> >
> > The performance of GATNE is clearly better in the case of intra-layer prediction. However, as discussed in the previous comment, inter-layer link prediction has several well-motivated applications ranging from recommendation systems to biology and knowledge graph analysis. For such applications, our model clearly outperforms competitors, yielding more reliable results.
> > Furthermore, if we consider the general behavior of the methods without distinguishing intra- and inter-layer connections, one can see that our algorithm is clearly the best solution. Such an analysis is reported in Table 4 of our revised paper.

---

### Official Review · Reviewer_FRGy · 2023-11-10

**Soundness:** 3 good
**Presentation:** 3 good
**Contribution:** 2 fair
**Rating:** 3
**Confidence:** 5

**Summary:**

Summary: The authors address the problem of intra and inter-layer link prediction in multiplex graphs. The authors mainly focus on the problem of inter-layer link prediction and show superior results on this task.

**Strengths:**

- The authors show interesting performance gain on inter-layer link predictions.
- The paper is clearly written.

**Weaknesses:**

- I’m not sure if the problem of inter-layer link prediction is important. I’m not aware of the importance of the problem. I know that there is a graph alignment problem, but I’m not sure whether the inter-layer link prediction problem needs to be treated separately especially when it comes at the cost of performance on the intra-layer link prediction
- Multi-relational data can be modeled as heterogeneous and multiplex models even when different instances of the same nodes have different features. In this paper, the authors only compare multiplex methods and do not compare with heterogeneous methods, which are more popular. In the heterogeneous case, it will be about predicting different types of self-loop. Also, evaluate your model on the Heterogeneous benchmark.
References
[1] Are we really making much progress? Revisiting, benchmarking, and refining heterogeneous graph neural networks
[2] Revisiting Link Prediction on Heterogeneous Graphs with a Multi-view Perspective

**Questions:**

See weaknesses

---

> ### Author Response · Authors · 2023-11-22
> **Reviewer 1 comments' answers**
>
> **Question 1.**
>
> ***Author answer:***
>
> The inter-layer link prediction problem, which is closely related to graph matching (Tang et al., 2022), finds many crucial applications such as: i) in online social network (OSN) analysis, the linkage of user identities across different OSNs, an emerging task in social media that has attracted increasing attention in recent years (Shu et al., 2017). User identity linkage finds potential impact in different domains, from recommendation systems to cybersecurity (Tang et al., 2020); ii) identifying the same genes or proteins across different biological networks, such as gene expression, protein interaction, metabolic pathways, etc. This can help to discover the molecular mechanisms of diseases and to find potential drug targets (Jain et al., 2023); iii) matching the same entities across different knowledge graphs (Azmy et al., 2019). This can help to enrich the semantic information and to improve the query answering and reasoning capabilities.
> The range of possible applications of inter-layer link prediction ultimately motivates the development of embedding algorithms for multiplex networks that are able to distinguish intra-layer and inter-layer links and to reconstruct both connectivity patterns.
>
> ---
> **Question 2.**
>
> ***Author answer:***
>
> We appreciate the reviewer's suggestion of using self-loops, but they may not reflect the information in the multiplex model. Self-loops could indicate information that is easy to obtain in multi-relational graphs. For example if you have a multi-relational graph with three different types of edges (red, green, yellow) you can easily put three self-loops in each node (red-green, red-yellow, green-yellow). This comes from the fact that using a multi-graph representation for multi-relational data is based on the assumption of knowing which nodes are linked, namely the identity of each node in every layer of relationships.
> Therefore, link prediction in heterogeneous graphs cannot be directly used for predicting the inter-layer connections, because heterogeneous graphs have different types of nodes and edges, while multiplex networks have the same type of nodes but different types of edges. Link prediction in heterogeneous graphs requires to consider the node and edge attributes and semantics, while inter-layer prediction requires to consider the layer and network properties. Therefore, different methods and features are needed for these two tasks.

---

> ### Comment · Reviewer_FRGy · 2023-11-23
> **Reponse**
>
> 1> Thank you for highlighting a few applications of inter-layer link prediction. Adding experimental results that compare a modified version of these methods to multiplex networks or comparing these methods with a simple graph version of a multiplex network will strengthen the paper.
>
> 2> I'd like to clarify that I was only referring to using heterogeneous graph models as baselines and not using heterogeneous datasets for evaluation.
> Let's say you have the following multi-layer graph with edges (node u, node v),
> - Layer 1: {{v11,v12}, {v11,13}}
> - Layer 2: {{v21, v22}}
> - Inter-layer: {v11,v21}
>
> This can be converted a heterogeneous graph with edges (node u, node v, relation type r)
> - Edges: {{v11,v12,1}, {v11,13,1}, {v21,v22,2}, {v11,v21,1-2}}.
> Once converted, you can apply heterogeneous link prediction models to predict new edges of type 1-2.

---

### Meta-Review · Area_Chair_6S1S · 2023-12-07

**Metareview:**

In this submission, the authors proposed a multiplex graph embedding method called MPXGAT.
The proposed method is based on graph attention networks, considering the intra- and inter-layer connections among graph nodes jointly.
Compared with existing embedding methods for multiplex graphs, the proposed method shows its superiority to some degree.

Strengths: (1) The problem itself is important because multiplex graphs are common in real-world scenarios. (2) The tech route is reasonable to some degree, in my opinion.

Weaknesses: Reviewers have common concerns about the novelty of the proposed method and the clearness of the writing. Some reviewers also suggested more baselines.

**Justification For Why Not Higher Score:**

The technical contribution is incremental, and both writing and experiments should be enhanced further.

**Justification For Why Not Lower Score:**

N/A

---

### Decision · Program_Chairs · 2024-01-16

Reject